# Conservation of a flagship species: Health assessment of the pink land iguana, *Conolophus marthae*

**Giuliano Colosimo**[1,2☯], **Gabriele Gentile**[1☯], **Carlos A. Vera**[3], **Christian Sevilla**[3], **Glenn P. Gerber**[2], **Hans D. Westermeyer**[4], **Gregory A. Lewbart**[4,5,6] *

**1** Department of Biology, Tor Vergata University, Rome, Italy, **2** San Diego Zoo Wildlife Alliance, Escondido, CA, United States of America, **3** Technical Biodiversity Research, Dirección Parque Nacional Galápagos, Puerto Ayora, Galápagos, Ecuador, **4** College of Veterinary Medicine, North Carolina State University, Raleigh, NC, United States of America, **5** Colegio de Ciencias Biológicas y Ambientales, Universidad San Francisco de Quito USFQ, Quito, Ecuador, **6** Galápagos Science Center GSC, Isla San Cristobal, Galápagos, Ecuador

☯ These authors contributed equally to this work.
* greg_lewbart@ncsu.edu

**Data Availability Statement:** All relevant data are within the paper and its Supporting Information files.

## Abstract

The pink land iguana, *Conolophus marthae*, is one of four species of iguanas (three terrestrial and one marine) in the Galápagos Islands, and the only one listed as critically endangered by the IUCN. The species can only be found on the north-west slopes of the highest volcano on Isabela Island and was first described to science in 2009. As part of a population telemetry study, a health assessment was authorized by the Galápagos National Park. Wild adult iguanas were captured on Wolf Volcano in September 2019 and April 2021 to record morphological and physiological parameters including body temperature, heart rate, intraocular pressures, tear formation, and infrared iris images. Blood samples were also collected and analyzed. An i-STAT portable blood analyzer was used to obtain values for base excess in the extracellular fluid compartment (BEecf), glucose (Glu), hematocrit (HctPCV), hemoglobin (Hb), ionized calcium (iCa), partial pressure of carbon dioxide (pCO$_2$), partial pressure of oxygen (pO$_2$), percent oxygen saturation (sO$_2$%), pH, potassium (K), and sodium (Na). When possible, data were compared to previously published and available data for the other Galápagos iguanas. The results reported here provide baseline values that will be useful in detecting changes in health status among pink land iguanas affected by climate change, invasive species, anthropogenic threats, or natural disturbances. The collected data also provide an invaluable resource for conservation scientists planning to implement conservation strategies, like translocations, that may temporarily alter these baseline values.

## Introduction

The Galápagos pink land iguana, *Conolophus marthae*, was first sighted on Wolf Volcano (WV hereafter), Isabela Island, Galápagos, in 1986, but was not described to science as a separate and unique species until 2009 [1]. Despite its recent description, this species rapidly became a

**Funding:** GC received salary support from the San Diego Wildlife Alliance. GG received financial support, not salary, from the International Iguana Foundation and the Friends of Galápagos, Switzerland. The funders had no role in study design, data collection and analysis, decision to publish, or preparation of the manuscript.

**Competing interests:** The authors have declared that no competing interests exist.

conservation flagship, as it helped raising awareness about the value of biodiversity locally and internationally [2, 3]. The species is under the direct protection of Galápagos National Park Directorate (GNPD) and listed in Appendix II of the Convention on International Trade in Endangered Species of Wild Fauna and Flora [4]. It is classified as critically endangered by the IUCN Red List of Threatened Species [5]. Among the major threats for the species, we recognize: very small population size ($200 < N < 300$ adult individuals); small distribution area; lack of recruitment, with no hatchlings and very few juveniles observed since 2005; introduced alien species such as cats and rats. With such a small population, these animals are at risk of extinction, from both natural and anthropogenic events. Although information on the natural history and biology of *Conolophus marthae* have been accreting since its description [1, 3, 5–9], very little is known about the animal's overall physiology and baseline medical parameters. This called for an urgent action aimed at providing data concerning the relative health status for individuals of this species.

Peripheral blood biochemical, blood gas, and hematology parameters are useful for assessing lizards' health [10–14]. It is important to establish and publish species-specific baseline blood and other parameters for healthy individuals since disease, injury, pollutants, or starvation can result in blood value perturbations. As potential alteration of baseline values could also come from specific conservation actions, baseline data can be used to assess individuals' health status after a conservation action has been implemented [15, 16]. Several reference intervals have been established for iguanids and include: *Conolophus pallidus* and *C. subcristatus* [13], *Amblyrhynchus cristatus* [14], *Basiliscus plumifrons* [17], *Cyclura cychlura inornata* [18], *Cyclura ricordii* [19], and *Iguana iguana* [20–24]. To date, hematological parameters and base line values have never been recorded in *Conolophus marthae* iguanas. As part of a radio-satellite tagging ecology and population study, a health assessment study for the pink iguanas was authorized by the GNPD. A detailed description of the satellite tracking study is beyond the scopes of this article, and we refer to Loreti and colleagues for further information [25, 26]. Nevertheless, we took advantage of opportunity provided by this tracking study and proceeded in two ways: *i)* we sampled wild pink iguanas on WV, Isabela Island, in September 2019 and April 2021 to analyze blood chemistry and hematological parameters and establish an intra-specific baseline report of such parameters; *ii)* we collected all published blood chemistry data available for other Galápagos iguanas to perform, when possible, an inter-specific comparison of such parameters. For wild samples collected in 2019 and 2021 a complete veterinary health examination was performed. The examination included measurement of body temperature, heart rate, length, body weight, intraocular pressure, ocular tear production, collection of blood samples, ectoparasites, and, in some cases, feces. In the current study, we report on the blood chemical analysis and status of clinically healthy wild adult Galápagos pink land iguanas.

## Materials and methods

### Ethic statement

This health assessment was authorized by the Galápagos National Park Directorate (Permit # PC-04-21 issued to G. Gentile). The techniques used during this health assessment were also approved by the University of Rome "Tor Vergata" ethics and animal handling protocol and the San Diego Zoo Wildlife Alliance IACUC. All procedures were performed by and carried out under the supervision of a licensed veterinarian and author of this study (GL).

### Sampling procedure

We captured, tagged, and sampled 15 and 27 adult pink iguana individuals during field expeditions conducted in September 2019 and April 2021, respectively. Individuals were collected

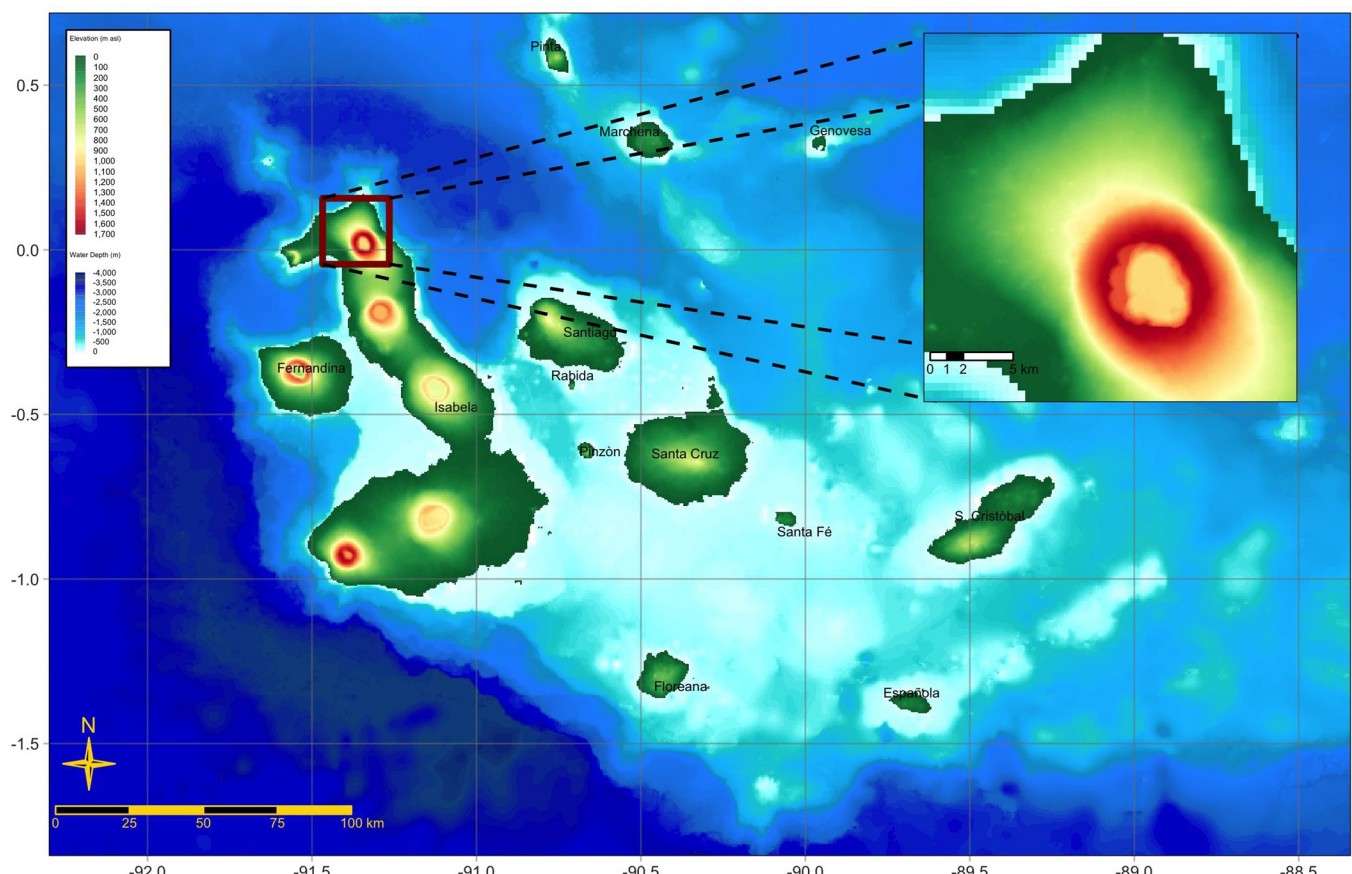

**Fig 1. Map of the Galápagos archipelago.** The archipelago is located ≈ 1000 km off mainland Ecuador. The inset shows a detail of the northwestern slopes of Wolf Volcano, the only site where pink iguanas are known to persist.

from an area on WV comprising approximately 6 km$^2$ at an altitude of 1600–1700 m (Fig 1). All iguanas were captured by hand or noose, either inside of or adjacent to their burrows. The animals were quickly transported to a field laboratory area (usually located within 10 m of the capture site) for blood drawing (usually within 5 min from capture). A blood sample of approximately 2.5 mL was obtained from the coccygeal hemal arch of each iguana using a heparinized 22-gauge 3.8 cm needle attached to a 10.0 mL syringe. This volume is considerably less than the safety threshold established for lizards (at 0.7 ml per 100 g [27]). The blood was divided into sub-samples and stored in a field cooler kept refrigerated using icepacks. Once the blood sample was secured the animal was examined, measured, weighed, and a custom designed GPS tracker was attached ([25, 26] for further details on the tracker devices).

Cloacal temperature and heart rate were recorded shortly after the blood was taken, and usually within few minutes from capture. Heart rate was recorded via a Doppler ultrasound probe (Parks Medical Electronics, Inc., Aloha, Oregon, USA) over the heart. An EBRO® Compact J/K/T/E thermocouple thermometer (model EW-91219-40; Cole-Parmer, Vernon Hills, Illinois, USA 60061) with a T-PVC epoxy-tipped 24 GA probe was used to determine core body temperature. Snout-vent length (SVL) and tail length (TL) were recorded with a flexible measuring tape and used to determine the total length of each individual. Body mass was measured with a digital scale (± 0.1 kg). The sex of the iguanas was determined by the presence or absence of hemipenes or by visually inspecting secondary sexual characteristics (presence/

absence of femoral pores, size of the individual, prominence of nuchal crest). Prior to release, ectoparasite load was determined by counting the number of ticks. For each animal, a sample of ticks was collected and preserved in 70% ethanol. Each animal was also scanned for the presence of a Passive-Integrated-Transponder (PIT) and checked for the presence of a brand. For never-before captured individuals a PIT was placed under the femoral skin of the right leg for long term identification and population monitoring.

A complete examination of the eyes was performed. Within 15 minutes of capture, internal ocular pressure (IOP) measurements were taken of the left (IOP_L) and right (IOP_R) eye using a rebound tonometer (TonoVet®, iCare, Tiolat, Helsinki, Finland). Intraocular pressures were measured on the Tonovet's® rebound tonometer on undefined patient setting (p). Disposable probes were used and changed between every patient. The tonometer was held in position perpendicular to the patient eye and approximately 5 mm from the corneal surface. Tear production was measured using the endodontic absorbent point paper test (EPPT) in iguanas sampled in 2019 but not for samples collected in 2021. Size 30, 40, and 45 endodontic paper points (Parallax® Veterinary Absorbent Paper Points) were used. The tapered end of the clean points was placed into the fornix in both the left and right eye. The paper was removed after 60 s and the length of moisture that was wicked on the point was measured with a millimeter ruler. External infrared photographs of each eye (Panasonic© Lumix DMC-ZS50 12.1MP) were obtained for each animal sampled in 2019 and reviewed for ocular abnormalities by a board-certified veterinary ophthalmologist (HDW). Before releasing the animal in the exact location where it was caught, a unique ID number was painted with non-toxic and washable paint to prevent recapture. The ID was painted on both flanks and the tip of the tail was also colored with non-toxic, washable white paint.

## Hematology parameters

The blood samples were used for measuring various hematological parameters. Approximately one drop was used for lactate analysis; approx. one drop for glucose analysis; about 0.1 mL was loaded into an i-STAT Clinical Analyzer (Heska Corporation, Fort Collins, Colorado, USA) utilizing Chem8 cartridges (see later in text for details); about 0.05 mL was used for centrifugation with a portable microcentrifuge (Eppendorf North America, Inc., centrifuge model 5424) 5 min. at 14,000 G to determine packed cell volume (PCV) and total solids (TS). The PCV was determined by measuring the percentage of cellular material compared to plasma in the tubes. Two drops of plasma were placed on a refractometer (Ade Advanced Optics, Oregon City, Oregon 97045, USA) and the total solids values recorded. We additionally used $\approx$ one drop of blood for making blood films on clean glass microscope slides (samples not yet analyzed). All the blood not used for hematological analyses was stored in 2% SDS lysis buffer in the field until long term storage in laboratory freezers.

The i-STAT Clinical Analyzer is a handheld, battery-powered, device that measures selected blood gas, biochemical, and hematology parameters using approx. 0.095 mL of non-coagulated whole blood. The following parameters were measured: base excess in the extracellular fluid compartment (BEecf), bicarbonate ($HCO_3^-$), glucose (Glu), hematocrit (HctPCV), hemoglobin (Hb), ionized calcium (iCa), partial pressure of carbon dioxide ($pCO_2$), total carbon dioxide ($tCO_2$), partial pressure of oxygen ($pO_2$), pH, potassium (K), and sodium (Na). The i-STAT automatically produces temperature corrected values for $pCO_2$, pH and $pO_2$ once the animal's body temperature is entered. Blood lactate (Lact.) was determined using a portable Lactate Plus™ analyzer (Nova Biomedical, Waltham, Massachusetts, 02454 USA). A glucometer (Accu-Check® Active, Roche) was sometimes used to obtain near instant glucose values in the field and compare them to the glucose values obtained by the i-STAT Clinical

Analyzer. To compare the validity and consistency of i-Stat readings, glucose, calcium, sodium, potassium and hematocrit parameters were measured using two different i-Stat cartridge types (Chem8 and CG-8) in samples collected in 2019. Readings were compared using a Wilcoxon Rank Sum test.

## Data compiling procedure

We reviewed available and published health and hematological data from other Galápagos iguanas. Data were sourced primarily from two publications by Lewbart and colleagues [13, 14] from individuals of *C. pallidus*, *C. subcristatus*, the two other species of Galápagos land iguanas, and *A. cristatus*, the Galápagos marine iguana. We were able to compile hematological and morphological data from 28 *A. cristatus* (four females and 24 males), 21 *C. pallidus* (12 females and nine males) and 30 *C. subcristatus* (19 females and 11 males). The protocols adopted while sampling individuals of these other species are the same as the ones adopted for this study, in fact they were performed by the same author (GL). For this reason, when possible, the data were used to perform inter-specific comparisons.

## Data analysis

Unless otherwise stated, all statistical analyses and data manipulation were performed in R v4.0.5 [28]. We first built a series of boxplots to identify potential outlier values in our data. While it is hard to imagine a scenario where outliers exist when looking at morphological and physiological characteristics, it is possible that mistakes while transcribing data in the field have occurred. Also, considering the relatively small sample size we had at our hand, we wanted to avoid outlier values to potentially influence intra and inter-specific comparisons. Outliers in all measured parameters were identified by looking at values falling outside of the inter-quartile range of data distribution and removed from further analyses (S1 Fig).

We calculated summary statistics (mean, standard deviation, minimum and maximum values) for all recorded parameters in males and females. For *C. marthae* samples, we report on the differences between sexes in all measurements considered using a Wilcoxon Rank Sum test. In pink iguanas we also used linear morphological features to calculate an estimate of individual's body condition as suggested by Peig and Green [29]. We first used the *smatr* R-package [30] to compute a scaling exponent (derived from a standardized major axis regression). This exponent was then used to calculate the Scaled Mass Index (SMI) for each individual using the following expression:

$$SMI = M_i \left[ {}^{L_0}/_{L_i} \right]^{b_{sma}}, \tag{Eq 1}$$

where $M_i$ and $L_i$ are the body mass and the linear body measurement of individual $i$ respectively; $b_{SMA}$ is the scaling exponent estimated by the SMA regression of M on L; $L_0$ is an arbitrary value of L (in this case we used the SVL arithmetic mean value for the study population). We accounted for the significant difference between sexes by computing two separate scaling exponents for different sexes prior to calculate the SMI for each individual. The SMI is a better predictor of individuals' health, as it should be an unbiased estimator of the overall body condition of the individual [29].

Considering the non-normal distribution of most measured variables and acknowledging for the reduced sample sizes, we used Wilcoxon Rank Sum test to compare all measured variables between species while also considering for sex differences. Significant differences were computed accounting for multiple comparisons using Bonferroni correction [citation needed]. The potential relation between morphological measurements (SVL, Mass, and SMI) and the

other parameters measured in pink iguanas was investigated using Spearman rank correlation coefficient ($r_s$). Linear regressions were calculated also between each measured biochemical parameter. Significance level was adjusted for multiple comparisons using Bonferroni correction.

## Results

All pink iguanas sampled were deemed clinically healthy based upon physical examination. We found no significant differences in measurements obtained using different i-Stat cartridges (S1 Table). Almost all iguanas (95%) collected between 2019 and 2021 had ticks (between 9 and 90). Ticks were collected and stored in 70% ethanol but remain in Galápagos for identification at a later time. No significant ocular abnormalities were noted on initial examination or on review of the infrared images. A representative image showing the detail captured in the photographs is presented in Fig 2. Overall, the intraocular pressure values for right (R) and left (L) eyes were similar and not significantly different ($W_{IOP}$ = 734.5, p-value = 0.42). The comparison between ocular pressure in left eyes of males and females produced a slightly significant result ($W_{IOP\_L\_m/f}$ = 276, p-value = 0.03; Table 1). Very similar and not significantly

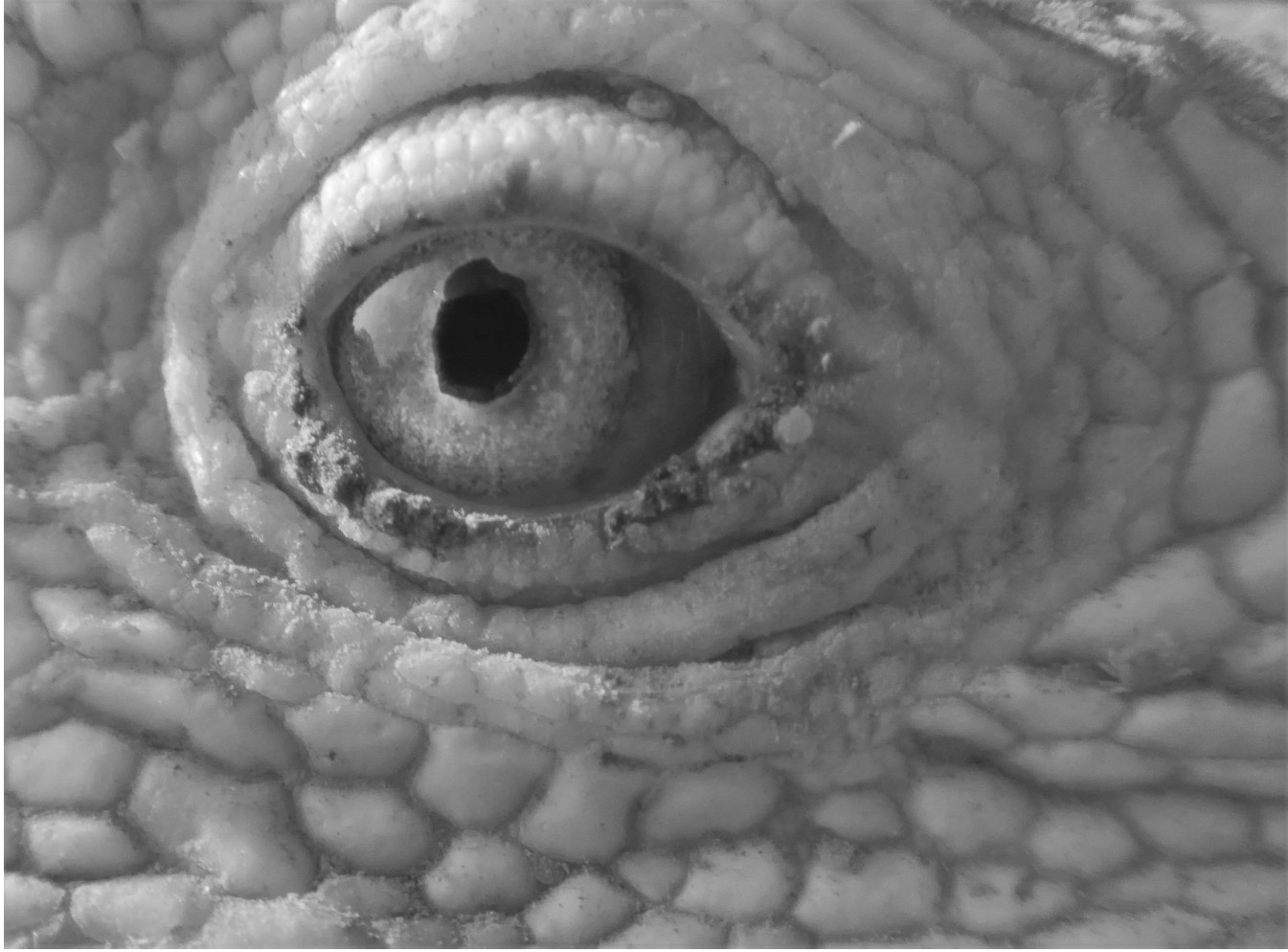

**Fig 2. Detail of infrared eye photography.** This image shows a detail of the infrared images collected and analyzed.

**Table 1.** *Conolophus marthae* intra-specific comparisons.

| | Males | | | | Females | | | | P |
|---|---|---|---|---|---|---|---|---|---|
| | Mean | Sd | Min/Max | n | Mean | Sd | Min/Max | n | |
| Svl (cm) | 46.36 | 3.65 | 37.40/53.00 | 17 | 41.13 | 4.03 | 33.60/50.20 | 22 | <<0.01 |
| Mass (Kg) | 5.36 | 0.64 | 4.40/6.50 | 16 | 3.76 | 0.82 | 2.20/4.87 | 23 | <<0.01 |
| HR (bpm) | 69.83 | 18.34 | 32.00/92.00 | 18 | 77.83 | 14.81 | 60.00/108.00 | 23 | 0.32 |
| RR (brpm) | 23.06 | 7.88 | 12.00/36.00 | 18 | 21.33 | 7.09 | 10.00/36.00 | 21 | 0.58 |
| T °C | 31.43 | 5.67 | 21.30/39.20 | 18 | 31.77 | 5.35 | 22.40/40.10 | 21 | 0.95 |
| IOP_R | 7.18 | 2.44 | 2.40/11.00 | 18 | 6.94 | 2.43 | 3.20/12.00 | 23 | 0.60 |
| IOP_L | 8.21 | 2.24 | 5.00/11.20 | 18 | 6.63 | 1.31 | 4.80/10.00 | 22 | 0.03 |
| EPPT_R | 15.13 | 6.12 | 4.50/22.50 | 8 | 13.08 | 3.97 | 7.50/17.00 | 6 | 0.39 |
| EPPT_L | 15.06 | 2.70 | 10.50/18.50 | 8 | 11.10 | 1.29 | 10.00/12.50 | 5 | 0.01 |
| BEecf | -2.71 | 1.50 | -4.00/0.00 | 7 | -7.14 | 5.90 | -17.00/-1.00 | 7 | 0.15 |
| Glu (mmol/L) | 154.47 | 22.41 | 110.00/192.00 | 17 | 155.22 | 26.19 | 110.00/209.00 | 23 | 0.90 |
| Hb (g/L) | 10.89 | 1.68 | 7.50/14.30 | 18 | 10.51 | 1.34 | 8.50/12.90 | 22 | 0.52 |
| $HCO_3^-$ (mmol/L) | 22.95 | 2.83 | 19.30/27.80 | 8 | 19.19 | 4.32 | 12.00/24.00 | 7 | 0.08 |
| HctPCV (%) | 32.12 | 5.01 | 22.00/42.00 | 17 | 30.95 | 4.10 | 23.00/38.00 | 22 | 0.58 |
| iCa (mmol/L) | 1.33 | 0.16 | 1.00/1.62 | 17 | 1.45 | 0.15 | 1.13/1.77 | 23 | 0.02 |
| K (mmol/L) | 3.64 | 0.85 | 2.00/4.40 | 7 | 3.13 | 1.20 | 2.00/5.30 | 7 | 0.30 |
| Lactate (mmol/L) | 9.19 | 4.46 | 3.40/18.10 | 18 | 10.25 | 4.35 | 1.90/17.10 | 23 | 0.39 |
| Na (mmol/L) | 160.82 | 3.94 | 154.00/168.00 | 17 | 155.09 | 3.32 | 150.00/161.00 | 22 | <<0.01 |
| $pCO_2$ (mmHg) | 27.58 | 7.48 | 13.10/35.60 | 8 | 29.88 | 1.33 | 28.30/31.80 | 5 | 1.00 |
| $pH_{37°C}$ | 7.40 | 0.11 | 7.23/7.52 | 8 | 7.31 | 0.14 | 7.14/7.47 | 7 | 0.23 |
| $pO_2$ (mmHg) | 45.25 | 18.95 | 20.00/78.00 | 8 | 43.00 | 14.12 | 26.00/57.00 | 7 | 0.86 |
| $tCO_2$ (mmHg) | 23.88 | 3.79 | 16.00/29.00 | 17 | 22.91 | 3.99 | 16.00/29.00 | 23 | 0.39 |
| TS (g/L) | 7.63 | 1.16 | 5.40/9.50 | 18 | 10.22 | 1.65 | 7.10/12.50 | 23 | <<0.01 |

Mean, standard deviation (Sd), minimum and maximum, number of individuals analyzed (n) of morphological features (Svl and Mass), general health parameters (Hearth rate–HR, Respiratory rate–RR, Temperature–T, Internal Ocular Pressure–IOP for right and left eyes, and tears production–EEPT for right and left eyes), and blood biochemical parameters (Base excess in extracellular fluid compartment–BEECF; Glucose–Glu; Hemoglobin–Hb; Bicarbonate–$HCO_3^-$; Hematocrit–HCT; ionized calcium–iCa; Potassium–K; Lactate; Sodium–Na; Partial pressure of carbon dioxide–$pCO_2$; pH measured at standardized temperature–$pH_{[37°C]}$; Partial pressure of oxygen–$pO_2$; Total carbon dioxide–$tCO_2$; Total protein solid–TS) measured in C. marthae samples and separated according to sex. The last column in this table (P) reports the P-value from a Wilcoxon Rank Sum test comparison.

different values were also recorded for overall tear production ($W_{EPPT}$ = 99.5, p-value = 0.69). As for IOP, we found a slightly significant difference in eye tear production between sexes in the left eye ($W_{EPPT\_L\_m/f}$ = 36.5, p-value = 0.01; Table 1 and S4 Fig). Overall, morphological characteristics of adult pink iguanas analyzed here are in lines with results from other studies (see for example [1, 3]), with males significantly larger and heavier than females ($W_{SVL}$ = 316.5, p-value << 0.001; $W_{Mass}$ = 356.5, p-value << 0.001). Scaled mass index ranged from 4.40–6.24 in males and 2.21–4.65 in females and no outlier values were identified, suggesting all adults are relatively healthy (Fig 3). A detail of all biochemical parameters recorded in pink iguanas is reported in Table 1. We found a significant difference between males and females only in ionized calcium values ($W_{iCa}$ = 113, p-value = 0.02), sodium ($W_{Na}$ = 324.5, p-value << 0.01), and total solids ($W_{TS}$ = 52, p-value << 0.01). In *C. marthae*, heart rate was positively correlated with body temperature ($r_s$ = 0.61, p-value << 0.001), while respiratory rate was positively correlated with lactate values ($r_s$ = 0.487, p-value = 0.001). Values of HctPCV were very strongly correlated with hemoglobin ($r_s$ = 0.98, p-value << 0.001). An overview of other significant correlations is presented in S5 Fig.

**Scaled Mass Index**

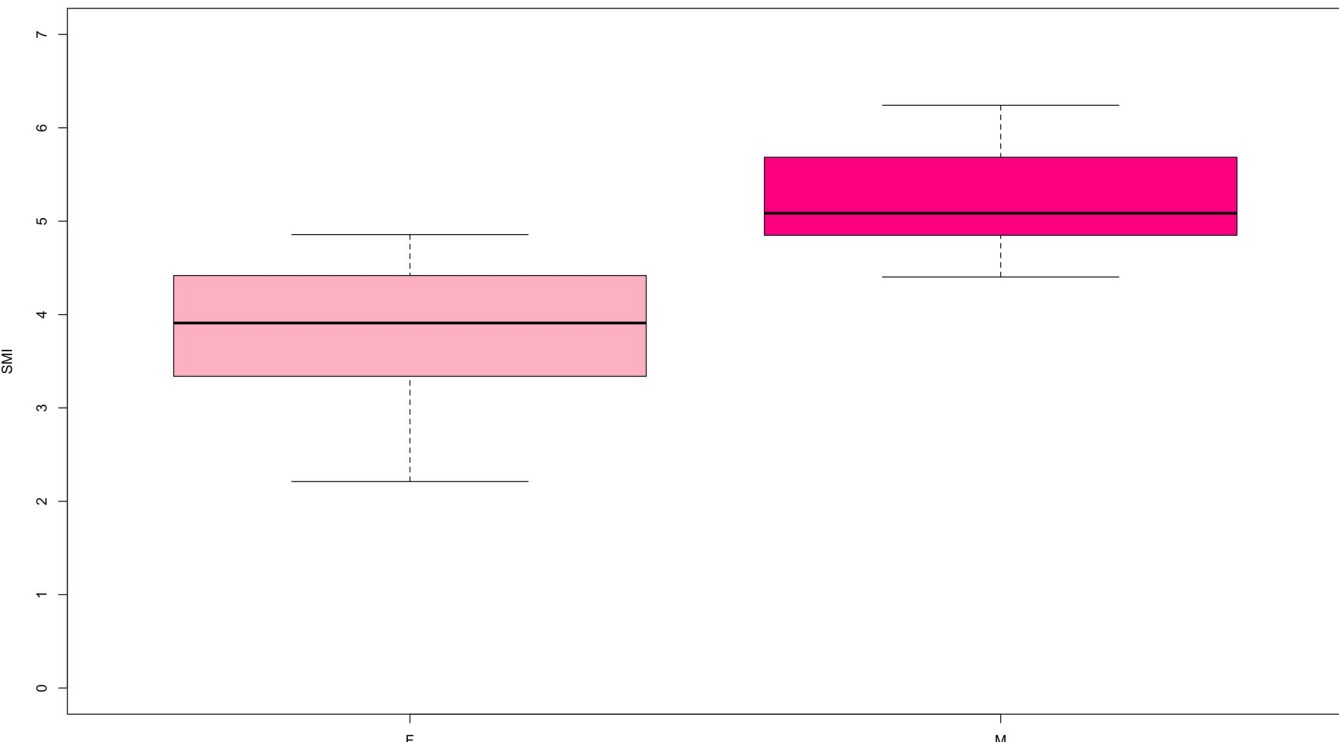

**Fig 3. Scaled *mass index* comparison.** Boxplots showing the distribution of SMI values in males and females of analyzed pink iguanas.

For a graphical comparison of all measured parameters across species see supplementary materials (S1–S4 Figs). Based on our pairwise Wilcox test we found two variables, ionized calcium and potassium, that did not differ in any of the considered groups (Fig 4). Values of sodium were found significantly differing between *A. cristatus* (both males and females) and all other groups considered (Fig 4). Similarly, the respiratory rate of *A. cristatus* male individuals was different from every other group but from that of *A. cristatus* females and *C. subcristatus* males (Fig 4).

## Discussion

The pink iguanas we examined were alert, robust, and clinically healthy. Despite relatively high tick and hemogregarine burdens physical examination and diagnostic test results support our good health assessment. Hemogregarines are known to affect this species [31], and does not appear to produce any clinical signs, such as lethargy, open mouth breathing, weight loss, or dehydration that may be observed in immunocompromised animals [32]. In general, even though the *C. subcristatus* from Wolf volcano did show significant alteration in some measures of immune function, significant correlation between corticosteroid levels (or body condition index) and the number of ticks or parasitemia were not found in *C. marthae* [31]. This supports the hypothesis that, in *C. marthae*, ecto- and hemoparasites can be sufficiently tolerated and may have coevolved with their host, as is appears to be the case in other iguana species [33].

Most of the blood values we recorded for the pink land iguanas were similar to those reported in other iguanids. For example, the average PCV in green iguanas, *Iguana iguana*,

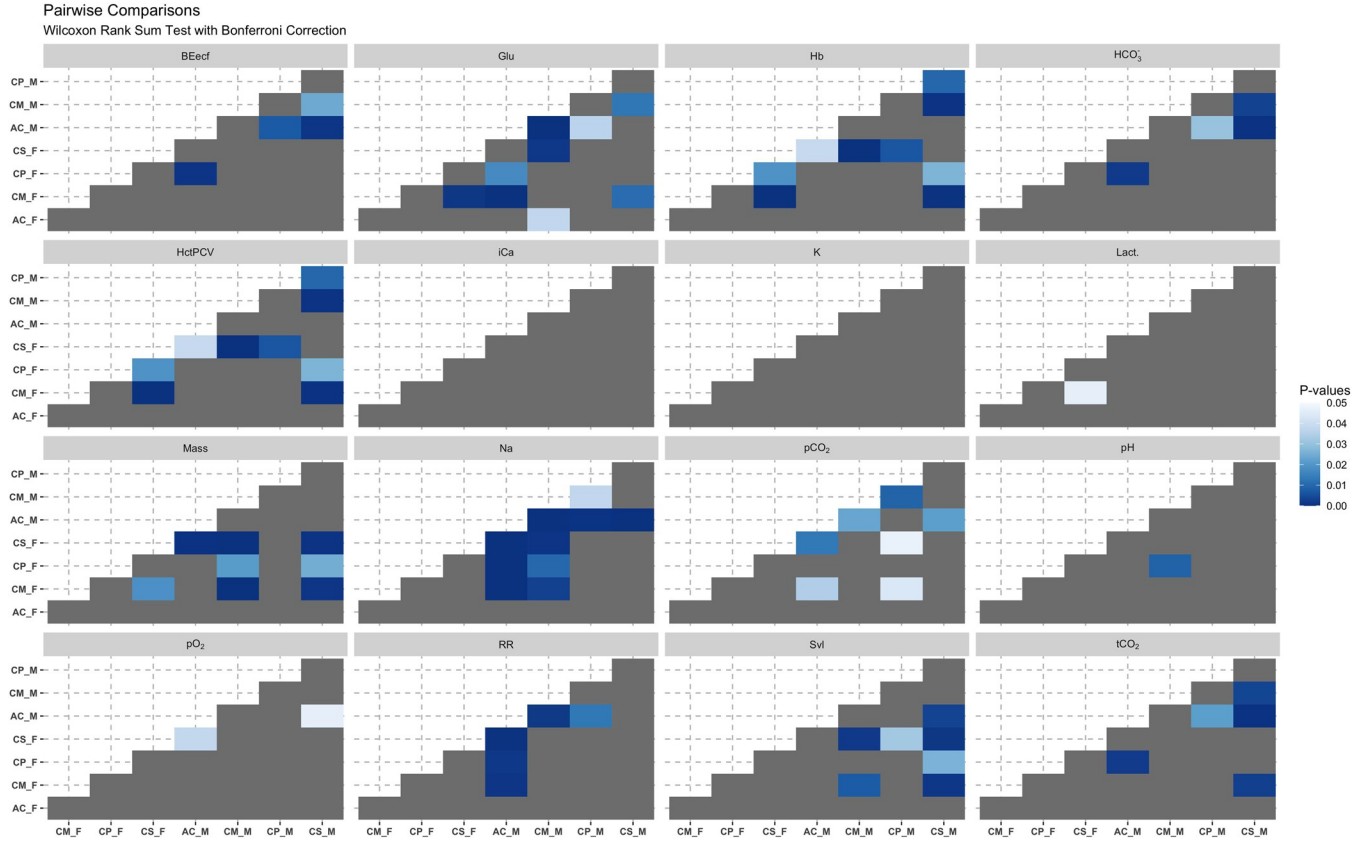

**Fig 4. Pairwise comparisons.** This figure shows the results Wilcoxon Rank Sum test pairwise comparisons between samples grouped based on species and sex. Darker shades of blue color indicate highly significant differences. Level of significance in pairwise tests has been adjusted using Bonferroni correction.

[22] was 36.7%, slightly higher than the average found in pink iguanas (31.75%). This value, indeed, is more similar to values reported for the basilisk lizard (*Basiliscus plumifrons*, 31.4%), and much closer to that of *C. subcristatus* [13, 17]. Despite this, for reptile species PCV values less than 18–20% would be considered anemic, and potentially associated with blood loss, chronic infections, malnutrition and exposure to toxins [27]. PCV values recorded in *C. marthae*, therefore, suggest that individuals are healthy. The blood sodium levels for green iguanas and basilisk lizards are 160 and 153.5 mmol/L respectively [17, 22], very similar and comparable to the average sodium levels for *C. marthae*, *C. subcristatus*, and *C. pallidus* are calculated at 156.9, 154.0, and 153.7 mmol/L respectively. Another general and easy way to determine health status and low stress is blood glucose. Basilisk lizards (that were held in cloth bags overnight and sampled 12 hours post-capture) had a fairly high level (203 mg/dL [17]) while *C. subcristatus* and *C. pallidus* had much lower, levels (126 and 135 mg/dL respectively [13]). The pink iguana glucose values, based on the i-STAT, average to 154.9 mg/dL, similar to that of captive green iguanas (166 mg/dL for males and 180 mg/dL for females [22]). Wild Allen Cays rock iguanas (*Cyclura cychlura inornata*) had a mean glucose of 189 mg/dL [18]. Glucose values are indeed affected by intrinsic as well as extrinsic factors and unless a large sample is available, it is hard to measure real hypo- or hyper-glycemia in reptiles [27]. Despite this, and after comparing the measured values with what is known from other species of iguanas, the authors consider the recorded values in pink iguanas clinically healthy.

Point-of-care analyzers like the i-STAT may require caution in interpretation, as published studies have found that some blood gas and hematocrit values are not always accurate or

reliable with certain non-mammalian species. In rainbow trout (*Onchyrhynchus mykiss*) results varied with temperature and only pH was a uniformly reliable value [34]. A study in sandbar sharks (*Carcharhinus plumbeus*) determined the i-STAT is not reliable for accurately measuring blood gases [35]. The i-STAT did not produce valid $sO_2$ or hemoglobin values in the bar-headed goose (*Anser indicus* [36]). After a comparison between the four different species of iguanas we noticed that the blood gas values and pH recorded were fairly consistent. The only notable exception is represented by the much higher sodium levels in marine iguanas [14]. This is most likely a result of the dietary and habitat difference between the terrestrial and marine iguanas. The marine iguanas also had higher potassium levels than all three land iguanas. The calcium values were comparable between all four species. Moreover, our comparison of parameters calculated using different i-Stat cartridges did not produce a significant difference (S1 Table) indicating that the values are indeed comparable.

In this study we document, for the first time, the base-line blood gas, biochemistry, and hematology values of the critically endangered and elusive *C. marthae*. The measured parameters, provide evidence indicating that the analyzed animals are healthy. Although we recognize that our sample size is not large enough to provide accurate reference intervals [10], our results represent a useful resource for veterinary scientists and other researchers especially considering that conservation strategies recently developed for this species call for actions, like translocation or/and head-start, that may temporarily alter these baseline parameters [37]. Moreover, although *C. marthae* individuals seem to be in good health at present, we recommend that the only existing population of this species is regularly monitored in order to be able to immediately track possible changes in health status that may be induced by the harsh and changing environment where the species lives.

## Supporting information

**S1 Fig.**
(DOCX)

**S2 Fig.**
(DOCX)

**S3 Fig.**
(DOCX)

**S4 Fig.**
(DOCX)

**S5 Fig.**
(DOCX)

**S1 Table. Comparison of measurements using different i-Stat cartridges.**
(DOCX)

## Acknowledgments

We are indebted to the park rangers of the Galápagos National Park for their invaluable support and friendship. This work is part of a long-term institutional agreement between the University Tor Vergata and the Galápagos National Park Directorate, aimed at the conservation of Galápagos iguanas. GAL thanks Diego Páez-Rosas, Juan Pablo Muñoz-Pérez, Carlos Mena, Stephen Walsh, and the Galápagos Science Center for their assistance and support.

## Author Contributions

**Conceptualization:** Giuliano Colosimo, Gabriele Gentile, Glenn P. Gerber, Hans D. Westermeyer, Gregory A. Lewbart.

**Data curation:** Giuliano Colosimo, Gabriele Gentile, Christian Sevilla, Hans D. Westermeyer, Gregory A. Lewbart.

**Formal analysis:** Giuliano Colosimo, Gabriele Gentile.

**Funding acquisition:** Giuliano Colosimo, Gabriele Gentile, Glenn P. Gerber, Gregory A. Lewbart.

**Investigation:** Giuliano Colosimo, Gabriele Gentile, Carlos A. Vera, Gregory A. Lewbart.

**Methodology:** Giuliano Colosimo, Gabriele Gentile, Carlos A. Vera, Glenn P. Gerber, Hans D. Westermeyer, Gregory A. Lewbart.

**Project administration:** Giuliano Colosimo, Gabriele Gentile, Carlos A. Vera, Christian Sevilla, Glenn P. Gerber.

**Resources:** Giuliano Colosimo, Gabriele Gentile, Carlos A. Vera, Christian Sevilla, Glenn P. Gerber, Hans D. Westermeyer, Gregory A. Lewbart.

**Software:** Giuliano Colosimo.

**Supervision:** Giuliano Colosimo, Gabriele Gentile, Glenn P. Gerber, Gregory A. Lewbart.

**Validation:** Giuliano Colosimo, Gabriele Gentile, Christian Sevilla, Hans D. Westermeyer, Gregory A. Lewbart.

**Visualization:** Giuliano Colosimo, Gabriele Gentile, Hans D. Westermeyer, Gregory A. Lewbart.

**Writing – original draft:** Giuliano Colosimo, Hans D. Westermeyer, Gregory A. Lewbart.

**Writing – review & editing:** Giuliano Colosimo, Gabriele Gentile, Carlos A. Vera, Christian Sevilla, Glenn P. Gerber, Hans D. Westermeyer, Gregory A. Lewbart.

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
