## [Decision Letter · Decision Letter 0]

10 Nov 2021

PONE-D-21-27173Health assessment of the pink land iguana, Conolophus marthae.PLOS ONE

Dear Dr. Lewbart,

Thank you for submitting your manuscript to PLOS ONE. Your work has been evaluated by 2 subject experts and the Academic Editor. All 3 view the work to be meritorious, but after careful consideration, it does not fully meet PLOS ONE’s publication criteria as it currently stands. Therefore, we invite you to submit a revised version of the manuscript that addresses the points raised during the review process.

REQUIRED CHANGES

 1. Remove redundant data displays. Several data displays are redundant to one another. For example, the data in Figures 1 and 3 are also presented in Tables 2 and 3. The authors need to decide on the best format for presenting data and only display the information once. Given that this paper is a reference collection for future research, the tables are likely better than the figures because they list specific values clearly.

 2. Add a study site figure. PlosOne has an international readership and it important to provide the geographic context for the work. A study site figure showing the location of the Galapagos Islands, Isabela, and the portion of the island studied, is required. 

 3. Allocate some data displays to supplementary online material (SOM). The authors present large collections of data, for example the correlation matrix in Figure 7, but dedicate almost no text in the manuscript to interpretation. If these datasets are not central to the main manuscript, they should be allocated to SOM. The authors need to examine carefully which figures belong in the main manuscript, and which ones should be removed and placed in SOM.

 4. Discussion should be reorganized. The Discussion begins with a short summary of the findings that does not provide any insight into the importance of, or context for the work. This paragraph contains no citations. The first paragraph needs to be re-written to lead the Discussion with a targeted assessment of the key finding(s) of the work that is appropriately referenced. The second paragraph of the Discussion is a cautionary one about the blood chemistry methods used. This should come later after the key biological findings have been discussed.  5. The authors should carefully consider the specific comments provided by the referees. 6. Data accessibility requirement is not met. he raw data for all aspects of the study need to be made publicly available. It is stated by the authors that it is all in the manuscript, but there are no raw data sets provided.

We look forward to receiving your revised manuscript.

Kind regards,

Christopher M. Somers

Academic Editor

PLOS ONE

Journal Requirements:

2. In your Methods section, please provide additional location information about your study area, including geographic coordinates for the data set if available.

This work is part of a long-term institutional agreement between the University Tor Vergata and the Galápagos National Park Directorate, 

aimed at the conservation of Galápagos iguanas. G.C. was supported by a Post-Doctoral Research Fellowship from the San Diego Zoo Wildlife Alliance funded by a donation from the Kenneth and Anne Griffin Foundation. G.G. was supported by 

grants from the International Iguana Foundation and from Friends of Galápagos, Switzerland.  GAL thanks Diego Páez-Rosas, Juan Pablo Muñoz-Pérez, Carlos Mena, Stephen Walsh, and the Galápagos Science Center for their assistance and support.

6. Please amend the manuscript submission data (via Edit Submission) to include author Hans Westermeyer.

Additional Editor Comments:

1. Introduction - paragraph 1 last sentence. The authors state that the pink land iguana is a "flagship" species. What is meant by this? How can a species that was only recently discovered by science and is little known to the broader world be a "flagship"?

2. Methods - the authors drew a blood sample of 2.5 ml, but don't mention that this is less than a threshold for safety based on % of total blood volume for the species. In addition, the authors seemed to use very little of this 2.5 ml for their analyses. What was the rationale for taking 2.5 ml?

3. Methods - why is tear production an important health parameter to measure, and why is it important to compare right and left eyes?

4. Methods - in the "Intra-Specific" section the authors mention removing outliers, but give no rationale for doing so, or the procedure for identifying outliers. In the case of SVL and mass, it is hard to imagine a scenario where true outliers can even exist. More information is required here.

5. Table 1 - I find the information content in this small table to be very low. It could likely be removed and the values summarized in the text.

6. Some might question the ethics of capturing and sampling blood from 40% of the individuals in a species that exists nowhere else in the world. Did any of the animals show signs of stress or capture injury? Is baseline data of the kind collected really worth it? Some additional text in the Discussion addressing this point would be useful. 

Reviewers' comments:

Reviewer's Responses to Questions

**Comments to the Author**

1. Is the manuscript technically sound, and do the data support the conclusions?

Reviewer #1: Yes

Reviewer #2: Yes

2. Has the statistical analysis been performed appropriately and rigorously? 

Reviewer #1: Yes

Reviewer #2: Yes

3. Have the authors made all data underlying the findings in their manuscript fully available?

Reviewer #1: Yes

Reviewer #2: Yes

4. Is the manuscript presented in an intelligible fashion and written in standard English?

Reviewer #1: Yes

Reviewer #2: Yes

5. Review Comments to the Author

Reviewer #1: This is a well-written manuscript outlining field-collected health data from apparently healthy individuals of an endangered species of iguana from the Galapagos islands. The health data collected is thorough and well presented.

There are too many figures with multiple figures demonstrating the same morphometric and clinical data. The number of figures could be reduced to 3 or 4 and the rest of the material submitted as supplemental information. The gross images should be collated into a single image.

In the results section, it is written that "one sample escaped", but the authors likely mean that the iguana escaped and an incomplete set of samples was taken.

More information could be added to the discussion that there are limitations in interpreting clinical data from a small sample of animals as reference intervals could not be obtained from only 15 individuals. The difficulties in interpretation from an iSTAT field analyzer can be explored with paired analyses from an in-lab analyzer, but that was not pursued in this case and this could be addressed by the authors.

Although there is a small sample size and there could be issues with interpretation of the results of a field analyzer, the information is still valuable as baseline data for use in potential captive breeding and reintroduction as well as captive collections (are there individuals of this species kept in zoological collections at the moment?). The information regarding the presence of hemogregarine parasites without an overall negative impact on health is an important take-home from this manuscript and could be highlighted more in the discussion and abstract as it it likely associated with co-evolution.

Reviewer #2: Really important work for an endangered species. Technical language, and easy to understand for the reader. Results presented in a kind graphical way. The authors developed a vigorous statistic analysis.

1) Abstract should highlight the main results and/or important points of discussion.

2) Should consider to include some extra results of descriptive/summary statistics for hematological/biochemistry results (range, min, max, maybe quartiles)

3) For results tables: important to clarify if all the variables have the same number of individuals analyzed, or if not, include an "n" data column in the tables.

4) Could define if there are "local seasons" that externally influence the variables or if it's not taken into account because there're no significant weather variations or according to other analytic factors

5) Did the authors measure the total proteins on in-house equipment to verify or compare the field refractometry results of total solids (TS)?

6) "Data compiling procedure" section could be included in the first methodology heading and so diminish the amount of text.

7) Data of re-sampled individuals is included as separated data (2 samples per individual)? any chance the authors developed a statistical comparison of intraindividual variations between the two samplings?

8) The "heavy mite-infested" individual was included in the general "n" for data presentation?

9) Maybe include a table to summarize the opthalmological results, so it does not get lost in the text.

10) Finally, did the authors make statistical correlations between the grade of parasitism (external ("tick charge") or hemoparasites) and hematological/biochemical analytes? beyond the previous report's explanation made on the discussion...

6. PLOS authors have the option to publish the peer review history of their article (what does this mean?). If published, this will include your full peer review and any attached files.

Reviewer #1: No

Reviewer #2: **Yes: **Steven Barajas-Valero

---

## [Author Response · Author response to Decision Letter 0]

8 Feb 2022

Response To Reviewers

6 December, 2021

PONE-D-21-27173

Dear Dr. Lewbart,

Thank you for submitting your manuscript to PLOS ONE. Your work has been evaluated by 2 subject experts and the Academic Editor. All 3 view the work to be meritorious, but after careful consideration, it does not fully meet PLOS ONE’s publication criteria as it currently stands. Therefore, we invite you to submit a revised version of the manuscript that addresses the points raised during the review process.

REQUIRED CHANGES

1. Remove redundant data displays. Several data displays are redundant to one another. For example, the data in Figures 1 and 3 are also presented in Tables 2 and 3. The authors need to decide on the best format for presenting data and only display the information once. Given that this paper is a reference collection for future research, the tables are likely better than the figures because they list specific values clearly.

Answer: Thank you. We have removed Figure 1 but kept Figure 2 as it shows values (like IOP and EPP) not currently present in tables. For a detailed reference of the values, which we agree is very important, we decided to maintain the tables as well, but move them to the Supplementary online materials.

2. Add a study site figure. PLOS ONE has an international readership and it important to provide the geographic context for the work. A study site figure showing the location of the Galapagos Islands, Isabela, and the portion of the island studied, is required.

Answer: Thank you. We have now added a map of the Galápagos Islands with an inset focusing on Wolf Volcano, where the only population of pink iguanas is located. It is now the new Figure 1.

3. Allocate some data displays to supplementary online material (SOM). The authors present large collections of data, for example the correlation matrix in Figure 7, but dedicate almost no text in the manuscript to interpretation. If these datasets are not central to the main manuscript, they should be allocated to SOM. The authors need to examine carefully which figures belong in the main manuscript, and which ones should be removed and placed in SOM.

Answer: Thank you. We agree that there were too many graphics. We have now moved some of them (like figures previously labelled as 1 and 7) to the SOM. We also made sure to discuss the content of each graphic a little bit more, even for the ones moved to the SOM.

4. Discussion should be reorganized. The Discussion begins with a short summary of the findings that does not provide any insight into the importance of, or context for the work. This paragraph contains no citations. The first paragraph needs to be re-written to lead the Discussion with a targeted assessment of the key finding(s) of the work that is appropriately referenced. The second paragraph of the Discussion is a cautionary one about the blood chemistry methods used. This should come later after the key biological findings have been discussed.

Answer: Thank you. We have now reorganized the Discussion to according to the editor and reviewer suggestions.

5. The authors should carefully consider the specific comments provided by the referees.

Answer: All comments have been considered carefully and addressed using the track change feature in word.

6. Data accessibility requirement is not met. The raw data for all aspects of the study need to be made publicly available. It is stated by the authors that it is all in the manuscript, but there are no raw data sets provided.

Answer: We have prepared an Excel spreadsheet to share the raw data analyzed in this manuscript.

We look forward to receiving your revised manuscript.

 Kind regards,

 Christopher M. Somers

Academic Editor

PLOS ONE

Journal Requirements:

1. Please ensure that your manuscript meets PLOS ONE's style requirements, including those for file naming. The PLOS ONE style templates can be found at:

Answer: Thank you. We have now updated all the information and made sure that the manuscript meets PLOS ONE's style requirements.

2. In your Methods section, please provide additional location information about your study area, including geographic coordinates for the data set if available.

Answer: Thank you. We have added a map to give a better idea of the study location.

Answer: Thank you. We have now updated the information and provided details re the funds used for this study.

This work is part of a long-term institutional agreement between the University Tor Vergata and the Galápagos National Park Directorate, aimed at the conservation of Galápagos iguanas. G.C. was supported by a Post-Doctoral Research Fellowship from the San Diego Zoo Wildlife Alliance funded by a donation from the Kenneth and Anne Griffin Foundation. G.G. was supported by grants from the International Iguana Foundation and from Friends of Galápagos, Switzerland. GAL thanks Diego Páez-Rosas, Juan Pablo Muñoz-Pérez, Carlos Mena, Stephen Walsh, and the Galápagos Science Center for their assistance and support.

Answer: Thank you. We have now changed the information in the manuscript and we will make sure to provide the correct funding statement during the resubmission process.

Answer: Thank you. We have submitted the dataset to Dryad servers and we have DOI that is provided in the supplementary on line material for full access to the data used in this article.

6. Please amend the manuscript submission data (via Edit Submission) to include author Hans Westermeyer.

Answer: Thank you. We have now added the author.

Additional Editor Comments:

1. Introduction - paragraph 1 last sentence. The authors state that the pink land iguana is a "flagship" species. What is meant by this? How can a species that was only recently discovered by science and is little known to the broader world be a "flagship"?

Answer: We have clarified this issue by modifying the sentence as follow: “Despite its recent description, this species rapidly became a flagship species, as it helped raising awareness about the value of biodiversity locally and internationally [2,3].”

2. Methods - the authors drew a blood sample of 2.5 ml, but don't mention that this is less than a threshold for safety based on % of total blood volume for the species. In addition, the authors seemed to use very little of this 2.5 ml for their analyses. What was the rationale for taking 2.5 ml?

Answer: The safety threshold considered for lizards is 0.7 mL of blood per 100 g of mass. The amount of blood sampled for our study is considerably lower. We have now clarified this in the manuscript as well, also providing the appropriate reference.

3. Methods - why is tear production an important health parameter to measure, and why is it important to compare right and left eyes?

Answer: The overall rationale to collect baseline data is to have a spectrum of useful parameters. While tear production per se may not be directly correlated with the health status of an individual, it nevertheless represents a parameter that can contribute to the health assessment. For example, should an individual be found with an extremely high tear production it would be interesting to investigate the causes of such alteration, which in turn could lead to the discovery of a threatening condition.

4. Methods - in the "Intra-Specific" section the authors mention removing outliers, but give no rationale for doing so, or the procedure for identifying outliers. In the case of SVL and mass, it is hard to imagine a scenario where true outliers can even exist. More information is required here.

Answer: We have now clarified the issue and explained how and why we looked for outlier values in our dataset.

5. Table 1 - I find the information content in this small table to be very low. It could likely be removed, and the values summarized in the text.

Answer: We have removed Table 1 and summarized its content in the text.

6. Some might question the ethics of capturing and sampling blood from 40% of the individuals in a species that exists nowhere else in the world. Did any of the animals show signs of stress or capture injury? Is baseline data of the kind collected really worth it? Some additional text in the Discussion addressing this point would be useful.

Answer: Thank you for this comment. We indeed considered the implication of this kind of approach. All the procedures adopted in the field to collect individuals and take specific samples have been approved by all parties involved in this research (GNPD, SDZW, Tor Vergata, NCSU). We have now expanded the discussion to also address this issue. 

Reviewers' comments:

Reviewer's Responses to Questions

Comments to the Author

1. Is the manuscript technically sound, and do the data support the conclusions?

Reviewer #1: Yes

Reviewer #2: Yes

2. Has the statistical analysis been performed appropriately and rigorously? 

Reviewer #1: Yes

Reviewer #2: Yes

3. Have the authors made all data underlying the findings in their manuscript fully available?

Reviewer #1: Yes

Reviewer #2: Yes

4. Is the manuscript presented in an intelligible fashion and written in standard English?

Reviewer #1: Yes

Reviewer #2: Yes

5. Review Comments to the Author

Reviewer #1: This is a well-written manuscript outlining field-collected health data from apparently healthy individuals of an endangered species of iguana from the Galapagos islands. The health data collected is thorough and well presented.

There are too many figures with multiple figures demonstrating the same morphometric and clinical data. The number of figures could be reduced to 3 or 4 and the rest of the material submitted as supplemental information. The gross images should be collated into a single image.

In the results section, it is written that "one sample escaped", but the authors likely mean that the iguana escaped and an incomplete set of samples was taken.

More information could be added to the discussion that there are limitations in interpreting clinical data from a small sample of animals as reference intervals could not be obtained from only 15 individuals. The difficulties in interpretation from an iSTAT field analyzer can be explored with paired analyses from an in-lab analyzer, but that was not pursued in this case and this could be addressed by the authors.

Although there is a small sample size and there could be issues with interpretation of the results of a field analyzer, the information is still valuable as baseline data for use in potential captive breeding and reintroduction as well as captive collections (are there individuals of this species kept in zoological collections at the moment?). The information regarding the presence of hemogregarine parasites without an overall negative impact on health is an important take-home from this manuscript and could be highlighted more in the discussion and abstract as it it likely associated with co-evolution.

Reviewer #2: Really important work for an endangered species. Technical language, and easy to understand for the reader. Results presented in a kind graphical way. The authors developed a vigorous statistic analysis.

1) Abstract should highlight the main results and/or important points of discussion.

2) Should consider to include some extra results of descriptive/summary statistics for hematological/biochemistry results (range, min, max, maybe quartiles)

3) For results tables: important to clarify if all the variables have the same number of individuals analyzed, or if not, include an "n" data column in the tables.

4) Could define if there are "local seasons" that externally influence the variables or if it's not taken into account because there're no significant weather variations or according to other analytic factors

5) Did the authors measure the total proteins on in-house equipment to verify or compare the field refractometry results of total solids (TS)?

6) "Data compiling procedure" section could be included in the first methodology heading and so diminish the amount of text.

7) Data of re-sampled individuals is included as separated data (2 samples per individual)? any chance the authors developed a statistical comparison of intraindividual variations between the two samplings?

8) The "heavy mite-infested" individual was included in the general "n" for data presentation?

9) Maybe include a table to summarize the opthalmological results, so it does not get lost in the text.

10) Finally, did the authors make statistical correlations between the grade of parasitism (external ("tick charge") or hemoparasites) and hematological/biochemical analytes? beyond the previous report's explanation made on the discussion...

6. PLOS authors have the option to publish the peer review history of their article (what does this mean?). If published, this will include your full peer review and any attached files.

Do you want your identity to be public for this peer review? For information about this choice, including consent withdrawal, please see our Privacy Policy.

Reviewer #1: No

Reviewer #2: Yes: Steven Barajas-Valero

---

## [Editor Report · Decision Letter 1]

22 Feb 2022

Health assessment of the pink land iguana, Conolophus marthae.

PONE-D-21-27173R1

Dear Dr. Lewbart,

We’re pleased to inform you that your manuscript has been judged scientifically suitable for publication and will be formally accepted for publication once it meets all outstanding technical requirements.

Kind regards,

Christopher M. Somers

Academic Editor

PLOS ONE
---

## [Editor Report · Acceptance letter]

11 Mar 2022

PONE-D-21-27173R1 

Conservation of a flagship species: Health assessment of the pink land iguana, *Conolophus marthae*

Dear Dr. Lewbart:

I'm pleased to inform you that your manuscript has been deemed suitable for publication in PLOS ONE. Congratulations! Your manuscript is now with our production department. 

Kind regards, 

on behalf of

Dr. Christopher M. Somers 

Academic Editor

PLOS ONE